# Multi-Device Nutrition Control

**DOI:** 10.3390/s22072617

**Published:** 2022-03-29

**Authors:** Carlos A. S. Cunha, Rui P. Duarte

**Affiliations:** CISeD—Research Centre in Digital Services, Polytechnic Institute of Viseu, 3504-510 Viseu, Portugal; pduarte@estgv.ipv.pt

**Keywords:** precision nutrition, food plans, IoT, machine learning, food logging

## Abstract

Precision nutrition is a popular eHealth topic among several groups, such as athletes, people with dementia, rare diseases, diabetes, and overweight. Its implementation demands tight nutrition control, starting with nutritionists who build up food plans for specific groups or individuals. Each person then follows the food plan by preparing meals and logging all food and water intake. However, the discipline demanded to follow food plans and log food intake results in high dropout rates. This article presents the concepts, requirements, and architecture of a solution that assists the nutritionist in building up and revising food plans and the user following them. It does so by minimizing human–computer interaction by integrating the nutritionist and user systems and introducing off-the-shelf IoT devices in the system, such as temperature sensors, smartwatches, smartphones, and smart bottles. An interaction time analysis using the keystroke-level model provides a baseline for comparison in future work addressing both the use of machine learning and IoT devices to reduce the interaction effort of users.

## 1. Introduction

Disease caused by inappropriate diets is responsible for 11 million deaths and hundreds of millions of disability-adjusted life years [1]. The use of technology to support health (eHealth) opens an expansive landscape of opportunities. The emergence of a large set of smart devices capable of facilitating physiological data recording and other forms of recording the health status has potentiated many new eHealth applications. Mobile phones and smartwatches are among the devices with the most potential because of their ubiquity and sensor capabilities installed [2,3,4].

The importance of nutrition to health is unquestionable. However, the specificity of nutritional requirements for a person demands personalized nutrition control. Nutritional requirements lean on body parameters, genetic and epigenetic makeup, daily routines, and history of disease or allergies. Thus, health professionals (e.g., doctors and nutritionists) must intervene to keep food plans adequate for the target person. Nonetheless, the biggest challenge is not elaborating the food plan but instead is the follow-up. That includes keeping the food plan always present to the user, replacing unavailable or undesired foods, adjusting food quantities to exceptional energy consumption, and using logged food intake data to readjust future food plan revisions. Food intake logging, in particular, benefits from automation since it is time-consuming, and the discipline demanded by its operationalization leads to high dropout rates of food plan execution.

State-of-the-art approaches for automation food intake logging exploit the recognition of food and quantities in images [5,6] taken using the phone camera and unconventional intrusive devices to detect swallowing patterns associated with calories intake [7]. Notwithstanding the innovation inherent to these approaches, they suffer from measurement errors summing to the error introduced by food tables to quantify nutrients. Plus, these solutions still require some interaction (e.g., opening the application and taking pictures). A more realistic solution to reduce human interaction costs is integrating the nutritionist and user systems and resorting to off-the-shelf smart devices.

Smart devices are essential tools to enable the ubiquity of food plans by allowing their visualization anywhere. Plus, they act as a data-gathering mechanism for logging macronutrients, micronutrients, and hydration levels. These data feed into a nutritional model that can support the nutritionist (or other health professionals) adjusting the next food plan iteration.

This article presents the requirements and concepts of a solution covering the food plan life-cycle from its creation by the nutritionist to its visualization, adaptation, and logging of food intake by the person. It also discusses the system architecture and design by focusing on

Devices for food plan creation, visualization, and food logging (smartphones, smartwatches, and smart bottles).Devices for capturing relevant data for food plan adaption (e.g., energy consumption).Data integration mechanism.

The rest of this article is organized as follows. Section 2 presents the related work. Section 3 defines the problem addressed in this article and enumerates the requirements of a possible solution. Section 4 presents the concepts and formulas used in food plan creation. Section 5 describes the system architecture and implementation. Section 6 and Section 7 describe the scenarios where the system will be tested. Finally, Section 9 presents the conclusions.

## 2. Related Work

This paper addresses a multidisciplinary problem connecting several research areas, such as precision nutrition, Internet of Things (IoT), web technologies, and machine learning.

*Precision nutrition* is an eHealth research area that depends on the person’s characteristics to deliver nutritional advice [8]. One prominent research topic in this area is when advice is supported by machine learning models created from several sources of data—e.g., dietary intake (content and time), personal, genetics, nutrigenomics, activity tracking, metabolomics, and anthropometric. Food intake monitoring, in particular, provides a fundamental source of data to machine learning algorithms for creating adequate diet models. However, traditional food logging systems are intrusive, forcing users to change their routines. Hence, user interaction with the system makes this activity one of the main contributors to food plan execution dropouts.

Several approaches for automatic food intake logging have been proposed. Wearables are devices with high potential in healthcare [9], since they could automate the process of food intake logging. The results of their exploratory use in nutrition to reduce the burden of manual food intake logging are presented in [7]. The authors explored using a smart necklace that monitors vibrations in the neck and a throat microphone to classify eaten food into three food categories. The resultant models trained with data produced by these wearables revealed higher accuracy for the microphone when compared to the vibrations sensor. Notwithstanding the potential of wearables for automatic logging of food intake, they are still in their infancy, requiring development to reduce intrusiveness and achieve close to perfect accuracy.

Visual-based dietary assessment approaches represent another type of appealing solution that resorts to pictures to determine the intake of food nutrients. Lo et al. [5] explores deep learning view synthesis for the dietary assessment using images from any viewing angle and position. An unsupervised segmentation method identifies the food item, and a 3D image reconstruction estimates the portion size of food items. Despite the high accuracy of the approach, the results depend on depth images with separable and straightforward objects, notwithstanding typical dishes that may overlap several food items. Another work estimates food energy based on images using the generative adversarial network (GAN) architecture [6]. It resorts to a training-based system, which contrasts with approaches based on predefined geometric models which bound the evolution of models to food with known shapes. The authors’ approach provides visualization of how food energy estimation is spatially distributed across the image, enabling spatial error evaluation.

While visual food inference represents a promising research topic for automatic logging of food intake, its accuracy is still unacceptable for most applications. An alternative method for food logging is using speech-to-text conversion to reduce the user’s interaction effort required to introduce nutrient information in the software application. Speech2Health [10] allows recording of food intake through natural language. A user-acceptance study using Speech2Health has shown several advantages of a speech-based approach over text-based or image-based food intake recording. Nevertheless, even minor errors resulting from identifying food names and portion sizes from voice excerpts are unacceptable for generic use. Privacy represents another issue that speech-to-text introduces in public environments.

Most related work addresses the problem of automatic food intake monitoring. Instead of explicitly addressing that problem, we devised a holistic approach that depends on food plans created by nutritionists and followed by target users. By confirming meals or logging changes, these users produce data for feeding the feedback loop that approximates the food plan progressively to the actual user’s needs. The availability of a baseline plan and the use of intelligent devices to record hydration, temperature, and energy expenditure reduce user interaction effort. Additionally, machine learning is applied to user preferences modeling, helping nutritionists choose the best food for the plan.

## 3. Problem Definition

Nutrition is a topic that has received more attention in the last decades due to its potential for benefiting from advances in technology. The ubiquity of smartphones and the emergence of wearable devices has created the opportunity to gather data automatically and support the user in deciding the best food to eat at each meal.

Many smartphone apps provide features to log intake meals and present nutritional statistics. However, choosing the best food plan for an individual requires a professional analysis that considers their physical condition (e.g., fat mass, lean mass, and weight), clinical condition, and goals. Discarding the health practitioner from the process may lead to inadequate food plans and be dangerous for individuals with health issues. Fortunately, it is possible to use technology to reduce the manual effort needed to manage the food plan life-cycle. The problems solved by a holistic solution span over the nutritionist and user (person following the food plan) domains.

We specified the requirements for the user and nutritionist domains with the support of several experts, such as nutritionists and doctors from a private hospital. We scheduled several meetings with these experts in two different phases: (1) requirement analysis, with the support of high-definition interface prototypes, and (2) deliverable analysis, where we tested software increments within a limited group of people by creating appointments, food plans, and performing food logging. Section A.1 and Section A.2 in Appendix A present the use cases for each of these domains.

### 3.1. Nutritionist Domain

We identified the following requirements for the nutritionist domain:The nutritionist creates an appointment with the person’s data and all the parameters needed to obtain the nutrients required to build the food plan. The system should calculate the energy expenditure.The nutritionist creates the food plan aligned with the nutrition goals obtained from the appointment. The system should suggest food according to user preferences and goals.

Nutritionists gather several types of data in the course of the appointment, which allows determining the person energy expenditure (Section 4) and other metrics and goals that can further support decisions during food-plan-making.

Energy expenditure is the core metric for devising the food plan. It provides the calories further distributed between macronutrients (i.e., proteins, carbs, and lipids) as follows:(1)energy=α∗protein+γ∗carbs+β∗lipids

After providing the data required to calculate the energy expenditure to the system, the nutritionist defines values for α,γ, and β. These values represent the contribution ratio of each macronutrient to the energy expenditure, which is fixed to a specific day and distributed between meals.

The energy expenditure and its distribution between macronutrients and meals are dependent on the person’s profile. For example, athletes have an increased demand for energy compared with sedentary people, and distribution of nutrients needs to be adapted to specific days (e.g., carbohydrate intake before and after exercise to help restore suboptimal glycogen reserves).

Fiber, water, and micronutrients are essential food plan elements unrelated to energy expenditure. The nutritionist adjusts the quantity of each nutrient to the person’s goals and condition. For instance, during demanding physical activity, the person may need drinks with added sodium to replace electrolyte losses. On the other side, a person with the risk of high blood pressure would benefit from lowering sodium intake.

Food plan creation is time-consuming because it involves the combination of different types of food adequate to the person. That combination should fulfill the target energy expenditure and its distribution between macronutrients, and approximate the micronutrients specified for the food plan. As for selecting alternative food when the user follows the plan (user domain), the user preferences model also supports the nutritionist in choosing the food to be added to the plan. Here, the contribution of each food to the goals established for energy, macronutrients, and micronutrients represents a crucial input for the classifier.

The nutritionist needs to revise the food plan to adjust the energy and nutrients to the user goals, respecting the subsequent appointments. For example, suppose the user goal is not to reduce fat mass but increase muscle instead. In that case, the total energy intake specified for the plan must be reduced and, consequently, the proportion of macronutrients contributing to that energy. Since energy expenditure occupies the top of the energy breakdown hierarchy, it will drive food plan adaption according to data gathered during previous food plan executions. Smart devices may improve the accuracy of energy expenditure in further food plan revisions. The *physical activity energy expenditure* (Section 4.2) represents one component of energy expenditure that can be easily captured with acceptable accuracy by smartwatches (or fit bands), alone or combined with heart rate straps. These data combined with food and water intake logs—registered through the system interface or obtained through intelligent bottles—provide elements required to tune the successive food plan revisions.

### 3.2. User Domain

We identified the following requirements for the user domain:The person accesses the meals defined in the food plan for the current day or specific event using the mobile phone or smartwatch.The person confirms the ingestion of the meal as it is in the food plan.The person searches for alternatives to the current meal with equivalent nutrition characteristics aligned with their preferences model.The person logs other food eaten not present in the food plan.The smart bottle logs water ingestion with respect to a specific period.The smartwatch logs the calories spent by the person during the day with respect to physical activity.All logs are either associated with a specific day or to an event (e.g., sports practice).

Nutritionists must design food plans aligned with user conditions and preferences. Further, users demand ubiquitous food plan visualization and logging mechanisms with small interaction costs. While interaction efforts depend heavily on user interface design, off-the-shelf IoT devices can be valuable tools to reduce human interaction with the system. These devices may be balanced with efficient user interfaces to reduce food plan execution abandonment.

As food plans are fixed to days of the week, repeating for several weeks, users may often lack some ingredients when executing the plan. Hence, the system may suggest alternative food according to the nutritional equivalence and user preferences—using historical data for similar meals, days of the week, months, or even weather contexts.

### 3.3. Automation Limits

The number of interactions with the system and the individual interaction cost determine the total user interaction effort. Logging of meals intake as in the food plan requires a small interaction effort since the only input is the user confirmation in either the smartphone or smartwatch. Sometimes that happens in batches (e.g., by the end of the day), resulting in low interaction costs and a small number of interactions (one per meal), as presented in Table 1. In this scenario, the user domain can benefit from integrating the food plan built by the nutritionist with the smartphone application that allows its visualization and confirmation of intake meals.

Water intake logging demands a higher number of user interactions when compared with meal confirmation. The user may take a sip of water dozens or hundreds of times a day to be hydrated. Consequently, water intake logging is more complex unless they stick to a standard behavior, such as drinking from the same bottle and logging the bottle storage capacity when they finish. However, even that standard method has flaws because the user may never finish the last bottle refill during the day or replace it with new water. Smart bottles may potentially reduce the number of user interactions for water intake logging since all the logged water intake is sent to the cloud service and made accessible to our system without user interaction.

While the previous scenarios offer an automation opportunity, some actions are difficult to automate, such as logging food not registered in the food plan. As shown in Table 1, notwithstanding the small number of interactions during the day, the interaction cost of individual actions is high—justified mainly by the search for additional food and the introduction of respective quantities. In addition, their automation is complex, and the closest state-of-the-art approaches rely on machine learning to identify food in pictures taken using the phone. However, these approaches are still far from one hundred percent of accuracy, which leads to large errors summed from

Errors resultant from the identification of food objects;Errors inherent to values presented in food nutrient composition tables;Food quantification errors, introduced either by visual approximation or predicted from the picture.

Reduction of interaction costs with respect to activities with low automation potential needs to be handled at the interface design level. The user application interface should be optimized to reduce the effort of food-searching for the *changing meal* and *add extra food* actions.

**Table 1 sensors-22-02617-t001:** Interaction effort of main actions for each device.

Action	Smartphone		Smartwatch		Smart Bottle	
	Interaction Cost	Number of Interactions	Interaction Cost	Number of Interactions	Interaction Cost	Number of Interactions
Meal confirmation	low	low	low	low	n/a	n/a
Changing meal	high	low	n/a	n/a	n/a	n/a
Add extra food	high	low	n/a	n/a	n/a	n/a
Water logging	low	high	low	high	none	none

## 4. Energy Expenditure

The user energy expenditure drives the creation of food plans. Adequate diets approximate intaken calories to the total energy expenditure, which includes the resting energy expenditure (REE), physical activity energy expenditure (PEE), and thermic effect of food (TEF).

CB (caloric balance) in the human body approximates the CC (caloric consumption) to the sum of PEE, REE, and TEF.
(2)CB=CC−PEE−REE−TEF

This section presents the calculation of PEE and REE. Notwithstanding the low contribution of the TEF (between 3% and 10%) to the total energy expenditure (TEE), it may have an impact on obesity. However, we do not handle it in this article due to its high measurement complexity [11] created by dependency on several other variables (e.g., measurement duration) [12].

### 4.1. Resting Energy Expenditure

REE is considered equivalent to the basal metabolic rate (BMR). BMR is the minimum number of calories required for basic functions at rest. On the other side, RMR is the number of calories our body burns while at rest. Despite both definitions slightly differing, the Harris–Benedict equation [13,14] can approximate REE or other equivalent equations presented in Table 2 for calculation of BMR.

### 4.2. Physical Activity Energy Expenditure

PEE calculation involves converting metabolic equivalents of activities to calories expended per minute (cal/min), based on body weight and the varying exercise intensities. The physical activity level (PAL) is an inexpensive and accurate method for calculation of PEE, based on the average values of 24 h of TEE and REE, as follows:(3)PAL=TEE/REE

The effect of gender does not interfere with PAL calculation because the BMR absorbs the gender difference in energy needs accentuated by the heavier weight of men.

A table that associates physical intensity lifestyles to PAL values (Table 3) can simplify PAL calculation. In that context, TEE is the result of multiplying REE by the PAL value associated with the person’s lifestyle category [15].

Another method for PAL calculation combines the time allocated to habitual activities and the energy cost of those activities (Table 4). In this case, PAL represents an energy requirement expressed as a multiple of 24-hour physical activity ratio (PAR). Here, PAR is a factor of BMR (PAR is 1 when there is no energy requirement above REE). Intuitively, the energy cost (PAR) is multiplied by the activity time to obtain PAL [15,16].

### 4.3. Distribution of Nutrients

The TEE estimate represents the total calories in the food selected for the food plan. TEE is then broken down into macronutrients complemented with micronutrients.

Macronutrients are typically specified in grams per kilo of body weight; such is the case of protein, carbohydrates, and fat (lipids). The exception is fibers that are specified in total grams. Water is frequently classified also as being a macronutrient [17]. However, water and fiber have zero calories, unlike protein, fat, and carbs. Notwithstanding that fibers do not usually count as calories in food plans, one type of fiber, named soluble fiber [18], may be absorbed by the organism and thus provide the body with calories.

**Table 2 sensors-22-02617-t002:** Metrics provided by the user (rows) and calculated by the application (columns).

		Body Composition			Basal Metabolic Rate				Obesity		
		Muscle Mass (Lee)	Fat-Free Mass	AEC Rate	Harris-Benedict [19]	Mifflin-St Jeor [20]	Katch-McArdle [21]	Cunningham [22]	Body Mass Index [23]	Evans 3SKF [24]	Withers [25]
body composition	weight		✓		✓	✓	(muscle mass)	(muscle mass)	✓		
	height	✓			✓	✓			✓		
	fat mass		✓								
	skeletal muscle										
	bone mass										
	body cell mass										
	bone mineral content										
	intracellular water			✓							
	extracellular water			✓							
	gender	✓			✓	✓				✓	
	age	✓			✓	✓					
	race	✓								✓	
girths	tight	✓									
	calf	✓									
	relaxed biceps										
	contracted biceps										
	waist										
	gluteus										
	chest										
	crural										
lean mass segments	left/right arm										
	trunk										
	left/right leg										
fat mass segments	left/right arm										
	trunk										
	left/right leg										
skinfold	corrected upper arm	✓									
	calf										✓
	biceps										✓
	triceps									✓	✓
	supraspinal										✓
	subscapular										✓
	chest										
	axila										
	iliac crest										
	abdomen									✓	✓
	thigh									✓	✓
level of fat	visceral fat										

Compared with macronutrients, the number of micronutrients is vast, and for that reason, nutritionists only select a few to be used as control metrics during food plan creation. From the conversation with several nutritionists, we have chosen iron, calcium, sodium, and magnesium, because of their transversality over several population groups. However, the selection of micronutrients depends always on the target population group (e.g., elderly, young people, and athletes).

**Table 3 sensors-22-02617-t003:** Classification of lifestyles according to physical intensity (PAL values).

Category	PAL
Sedentary or light activity lifestyle	1.40–1.69
Active or moderately active lifestyle	1.70–1.99
Vigorous or vigorously active lifestyle	2.00–2.40

**Table 4 sensors-22-02617-t004:** Total energy expenditure for a population group.

Activities	Time Allocation	PAR	Time × PAR	Mean PAL
Sleeping	6	1.0	6.0	
Personal Care (dressing, showering)	2	2.3	4.6	
Eating	2	1.5	3.0	
Walking without a load	2	3.2	6.4	
Sitting	4	1.5	6.0	
Cooking	2	2.1	4.2	
Household work	2	2.8	5.6	
Light leisure activities	2	1.4	2.8	
Driving car	2	2.0	4.0	
Total	**24**		**42.6**	**42.6/24 = 1.8**

## 5. Architecture and Implementation

This section presents the architecture and implementation of the solution proposed in this article, divided between two front-ends: *nutritionist front-end* and *user front-end*.

### 5.1. Nutritionist Front-End

The nutritionist front-end (Figure 1) implements two important concepts: *appointment* and *food plan*.

The appointment is the concept responsible for managing the energy expenditure—and its distribution throughout macronutrients—and micronutrients, as presented in Section 4. Moreover, to support user monitoring between appointments, it should present all historical data entailing previous food plans and energy distribution by day of the week, event, and meal type.

Monitoring of physical conditions frequently resorts to the person’s goals, specified in terms of:Weight.Body fat.Visceral fat.Fat-free mass.Muscle mass.Body mass index.Exercise performance.

Control and analysis of generic user goals depend on the previous metrics, although specific people groups may require other specific metrics; such is the case of groups with specific diseases that require the control of specific body parameters.

Other important appointment data required for food plan making include the following:Bowel function.Sleep quality, and wake up and sleeping times.Person’s race.Food likes and dislikes.Night shifts.Job.Lifestyle.Clinical conditions.Current water intake.

Nutritionists rely on the appointment of data for food plan creation. While adding new meals and foods to the food plan, the nutritionist can balance the food calories with target energy and nutrients. They can also visualize other relevant information gathered during the elaboration of appointments.

### 5.2. User Front-End

The user front-end (Figure 2a) uses the food plan as the basis for preparing meals, searching for alternative foods, monitoring consumption of water and calories during the day, and food logging. Food is presented on the plate (Figure 2b)—useful for elderly, people with vision impairment, or those that may find it difficult using mobile/smartphones with mobile devices—and in the list format.

Daily statistics (Figure 2c) are valuable assets for monitoring calories, macronutrients, micronutrients, and hydration during the day. These values are paired with target values defined by the nutritionist in the food plan.

Notwithstanding the small screen sizes of smartwatches, they are practical for presenting meals (Figure 3a), sending notifications, and logging food intake. They also present statistics regarding nutrients intake (Figure 3b) and hydration (Figure 3c).

### 5.3. Architecture

Figure 4 presents the solution architecture composed of four different interfaces. The nutritionist interacts with the system to create appointments and food plans using a web application. On the other side, the user visualizes the current food plan or logs food ingestion using a mobile phone or smartwatch.

#### 5.3.1. Web Applications

The mobile application is delivered as a PWA (progressive web application). PWAs represent a new class of applications alternative to traditional mobile phone apps, with several advantages over them. Instead of being developed to a specific platform (e.g., iOS or Android), they are built as a web application that can work offline and be installed on any smartphone. A previous study reported PWAs 157 times smaller than React Native-based interpreted apps and 43 times smaller than Ionic hybrid apps [26]. The Twitter PWA consumes less than 3% of the device storage space as compared to Twitter for Android [27], and the Ola PWA is 300 times smaller than their Android app [28]. Additionally, they are cross-platform, although current implementations may require adaptation between some browsers.

Both applications with respect to the user and nutritionist front-ends were developed in LitElement [29], a base class to create lightweight web components. Design of the user front-end for smartphones embraces the PWA principles [30] (e.g., web application installability, and offline usage).

#### 5.3.2. Smart Bottle

Water consumption is logged either by the user—using the smartphone or smartwatch—or automatically by a smart bottle. We tested several smart bottles and decided on the Hidratespark [31], justified by its mature API and good construction and usability of the bottle. Plus, it can be easily integrated with Fitbit [32], which is used as a gateway to retrieve data to the user’s back-end.

Water intake goals defined in the food plan are adjusted according to the environment temperature. Temperature sensors provide the inputs to make that adjustment according to the rules stated in the food plan.

#### 5.3.3. Smartwatch

As explained in Section 4, determining the energy expenditure of one person is one of the main challenges in the creation of a food plan. Modern smartwatches provide a good approximation of energy consumption during physical activity. They provide valuable information to be used by food plan revision activities, enabling correction of energy expenditure values predicted by traditional methods during follow-up appointments (Section 4.2). Pedometers and heartbeat monitors incorporated in devices provide a good approximation of calories burned data [33].

#### 5.3.4. Preference Learning

Exploring machine learning techniques on logged data makes it possible to help nutritionists model user food preferences. These techniques build up a recommendation system [34], based on food preference models, that supports the selection of food during food plan creation. That system will also allow proposing food alternatives to the person following the plan. That may occur when the food is unavailable, or the person prefers other equivalent food.

Reinforcement learning seems an adequate tool for applying preference learning to food recommendation [35]. Starting without knowledge, the agent helps the nutritionist to choose the food and quantity for the food plan without breaking the constraints imposed by the goals established for macronutrients and micronutrients. The agent accuracy improves with the feedback received from the nutritionist and the intake of food logged by the user. The same agent can help the user choose equivalent food and quantities when executing the plan based on learned preferences and goals of nutrients.

### 5.4. Security

Security is a complex and wideband problem. It spans the human-related processes and the system level (e.g., network and application). Human misconduct is in the origin of several security threats in eHealth systems [36]. Training people and auditing security procedures is a natural way of reducing the risk of threats occurrence. Coordination between developers, users, organizations, and government regulators represents another security flaw source [37].

In this work, we handle security at the system design level. E-Health systems contain data that are sensitive to confidentiality, integrity, and availability threats [38]. There are different types of data sensitiveness. Personal data are the most critical data under management; thus, ensuring the confidentiality of these data is of the utmost importance. Hence, we segregate the user data in the application and provide one feature to remove these data anytime without compromising their food plan while an anonymous entity. The latter offers less security risk when unrelated to the person.

The design of the nutritionist application allows deletion of the user’s personal data without compromising the food plan management features, as long as an *ID* can identify the user. The segregation of functionality and data between the user and nutritionist applications offers an additional protective barrier. The user application uses an application token to communicate with the nutritionist application, and the former does not store or handle personal data—an ID identifies the user.

As much as personal data, authentication credentials are sensitive data demanding theft protection. The HTTPS already ensures protocol-level privacy in the communication channel. Plus, the front-end encrypts passwords before transmitting them to the back-end, and they are then handled and stored in an encrypted form.

Feature-oriented access control constrains the access to features available on each web page. There are three profile types: nutritionists, administrators, and users.

Risk management models, such as the one presented in [39], may complement our system design. Additionally, other protection schemes against complex attacks [40] are orthogonal to our system and may also be used.

## 6. Case Study: Alzheimer’s

Alzheimer’s disease is a progressive loss of mental function, characterized by degeneration of brain tissue, including loss of nerve cells, accumulation of an abnormal protein, and development of neurofibrillary braids [41]. Alzheimer’s patients become dependent on others, even for the most basic tasks. Controlling feeding and hydrating for an Alzheimer’s patient is thus a crucial activity performed by the person who supports their daily routine, called the informal caregiver (IC).

Conditions of malnutrition, super nutrition, and dehydration are common in people with diseases causing dementia. The loss of autonomy also manifests itself in their inability to demonstrate food needs. Therefore, it is fundamental to support the nutritionist in the preparation and follow-up of a food plan aligned with the patient’s needs. Food plan monitoring is undoubtedly a process that demands much discipline from the IC and the ability to deal with possible circumstantial adaptations, such as replacing foods prescribed in the food plan with other equivalents or changing the quantity of water consumed as a function of ambient temperature.

This case study investigates the problem of creating and monitoring diet plans in patients with dementia—such as those with Alzheimer’s. It allows the creation of nutritional plans by the nutritionists and the follow-up of these plans by the ICs through a mobile app to significantly increase the patient’s quality of life. The app will send the IC notifications regarding proper nutrition and hydration in the due moment. It also controls hydration using the smart water bottle. In addition, the application will suggest alternatives to plan foods if they are unavailable or rejected by the patient. Another feature important for this group is the dynamic adaptation of water administration to the patient as a function of environmental conditions observed by temperature and humidity sensors. This feature is vital when the patient is unable to express thirstiness.

## 7. Case Study: Sports

The recent growth in the pursuit of sporting activities, motivated by a widespread increase in the perception of the importance of maintaining physical fitness, campaigns explicitly aimed at combating physical inactivity, and opportunities created by the revelation of lesser-known modalities, has brought forward fundamental questions such as the correct nutrition of the practitioners. Several institutions and individuals involved in physical activity have integrated these concerns into their scope, including nutritionists.

Food plan elaboration and monitoring present two main challenges: (1) obtaining the person’s biometric data, eating habits, and energy consumption, and (2) monitoring user food intake and providing dynamic adaptation of the food plan.

Sports nutrition is one of the most complex areas of nutrition. It requires observing a comprehensive set of metrics, encompassing the athlete’s physical aspects, physical activity, and eating habits. Fortunately, devices for measuring specific physical parameters represent a common practice among athletes. The creation of data repositories to help nutritionists build the plan is only possible by automatically integrating data collected by these devices with other data not directly observable—such as dietary habits and subjective metrics. These repositories also contain data that can help adapt the food plan at its execution stage. For example, variations in temperature or physical intensity may demand quick changes in individual energy or hydration needs. In these scenarios, the support system uses data collected by devices to dynamically adjust the food plan and send alerts to athletes to eat food or water at the right time.

## 8. Interaction Results

This section presents the human–computer interaction cost associated with typical user tasks to visualize the food plan and log food intake.

Traditional methods used in the usability evaluation of an interface fall into two categories: (1) subjective opinion of users and experts—mainly applying questionnaires [42] and inspection methods [43,44]—and (2) objective techniques such as rules [45], analytics modeling [46], and automated testing [47,48]. Notwithstanding that these approaches provide important tools to determine the usability of the user interface, there is both cost and time needed to implement user interaction evaluation with acceptable coverage, coupled with the need to use experts to compensate for the user’s faults.

### 8.1. Keystroke-Level Model

We applied the keystroke-level model (KLM) [49] to the user interface depicted in Figure 2, for testing the quality of the human–computer interaction and estimating the time spent in critical tasks. In this model, a unit task is defined with two parts: *task acquisition* and *task execution*. The total time to complete a unit task is given by $T_{task}=T_{acquire}+T_{execute}$.

At the execution level, KLM provides physical, mental, and response operators with predefined time values. These operators are defined by a letter and include *K* (keystroke ≈ 0.12 s), *P* (point ≈ 1.1 s), *H* (homing the hand(s) on the keyboard or other device ≈ 0.4 s), *D* (draw is measured in real time), *B* (button press ≈ 0.1 s), *M* (mental preparation for action ≈ 1.35 s), and *R* (system response, which is a parameter measured in real time). The execution time is the sum of the time for each of the operators from the final KLM string $T_{execute}=T_{K}+T_{P}+T_{H}+T_{D}+T_{B}+T_{M}+T_{R}$.

### 8.2. Interaction Results

Table 5 presents the time required to execute each application task. The KLM string generated is represented in the *sequence of operators* column and the respective time required to execute each task in the *estimated time* column. The task *“update food entries for train and competition”* allows the creation of periodic food requirements and is specific to the sports scenario. In contrast, the Alzheimer’s and the sports scenarios share the other tasks. The results are presented for the user application since we aim to reduce user abandonment motivated by interaction costs resultant from food logging activities.

As expected, results show that tasks that change the original food plan for logging purposes manifest higher interaction costs. Food plan visualization requires 1.2 or 2.3 s, depending on the UI view. Logging one meal by confirming the original food plan only requires 1.2 s. On the other hand, logging tasks regarding food intake not present in the food plan are costly. Each extra food added to the food plan requires 8.66 s of the user’s time.

Manual logging of water using the application requires 3.6 or 4.8 s, depending on the view. The adoption of smart bottles avoids that interaction, which may repeat dozens of times during the day.

The interaction time of tasks performed by the smartwatch (e.g., energy expenditure logging) is not presented in this section. Despite the automation of the data logging process, the user can not perform any equivalent task manually.

**Table 5 sensors-22-02617-t005:** Interaction results.

Actor	Tasks	Sub-Tasks	Sequence of Operators	Estimated Time (s)
User	Visualize food plan	Graphical representation of the meal	PB	1.20
		Composition of the meal (by food)	PBP	2.30
User	Log food intake (items of the sub-tasks column marked with * repeat in the meal)	Add new food to the meal *	PBPBMHKKKMHPBPB	8.66
		Add new extra food (snack between meals) *	PBPBMHKKKMHPBPB	8.66
		Remove food *	PBPBPB	3.60
		Specify percentage of food intake *	PBPBPBPB	4.80
		Change food plan food *	PBPBPB	3.60
		Confirm food intake from food plan with no changes *	PB	1.20
User	Log water intake	Through food plan	PBPBPB	3.60
		Through interaction menu	PBPBPBPB	4.80
Fitbit (bottle)	Update water intake		-	0.00
User	Visualize statistics		PBPB	1.20
System	Update food entries for train and competition		-	0.00
User	Change active food plan (train or competition)		PBPBPBPB	4.80
User	Connect watch API		PBPBPBMH42KMHPB	13.34
User	Provide consent to access Fitbit API		PBPBPBR	4.60

### 8.3. Analysis of Results

The observed results of human–computer interaction times pinpointed the tasks requiring improvement of interaction times. They provide a baseline for evaluating other interaction schemes and assessing the contribution of automation (e.g., using IoT devices) to the goals established in this article. The lower the interaction time, the lower the user discipline needed to maintain a food plan visualization and logging process, and the lower the user abandonment rate.

We designed the application to minimize human interaction with the support of UI experts. The most challenging tasks using a UI (those with more significant interaction times) require the search of new food manually. Although the interface implementation can still be questionable in terms of the specific design that may compromise the generalization of results, it is evident that there is little space for improvement when we need to perform a generic search for food using text.

Machine learning techniques are natural solutions to help reduce the time required for logging extra food in addition to—or in replacement of—those present in the food plan. As referred to in Section 2, there have been several attempts to recognize food objects in pictures taken with the mobile phone to reduce the burden of manually logging food. However, interaction is still required to take the picture, and an accuracy less than perfect could even increase the interaction time since the user would need to correct these data. The previous rationale leads to a different strategy for exploring machine learning for reducing interaction time. Creating a food preferences model customized to each user would likely lessen the food search interaction time considerably. By resorting to historical data and observable features (e.g., user location, day of the week, and weather), the system can anticipate the consumption of specific food. In that scenario, the interaction time would be equivalent to confirming a meal in the food plan.

## 9. Conclusions

This article unveils the concepts, requirements, and technologies needed to build a system that could support the nutritionist in creating food plans aligned with the individual profile. Further, it presents an architecture and software developed for smartphones (PWA) and smartwatches. The software furnishes food plan visualization logging of food and water intake, among other related features. It also integrates other devices, such as smart bottle technology and temperature sensors, to reduce human–computer interaction.

The availability of off-the-shelf devices has brought unprecedented ways of gathering data from physical phenomena without resorting to direct human–computer interaction. We propose an architecture that integrates the nutritionist back-office, the user application, and smart devices, focused on interaction cost reduction when users follow a food plan. We presented a baseline of the human interaction effort associated with several tasks pinpointing the most critical (expensive) operations. Such baseline sustains the evaluation of future machine learning and IoT approaches targeting the reduction of human interaction effort when completing critical operations.

As future work, we plan to explore machine learning techniques to reduce interaction times in two demanding user groups: Alzheimer’s patients and athletes. The Alzheimer’s group offers interaction challenges since several caretakers are elderly and have difficulties using apps or are not motivated to use apps as a data logging mechanism. On the other hand, athletes are very disciplined but need tight control of food intake before, during, and after physical activity.

## Figures and Tables

**Figure 1 sensors-22-02617-f001:**
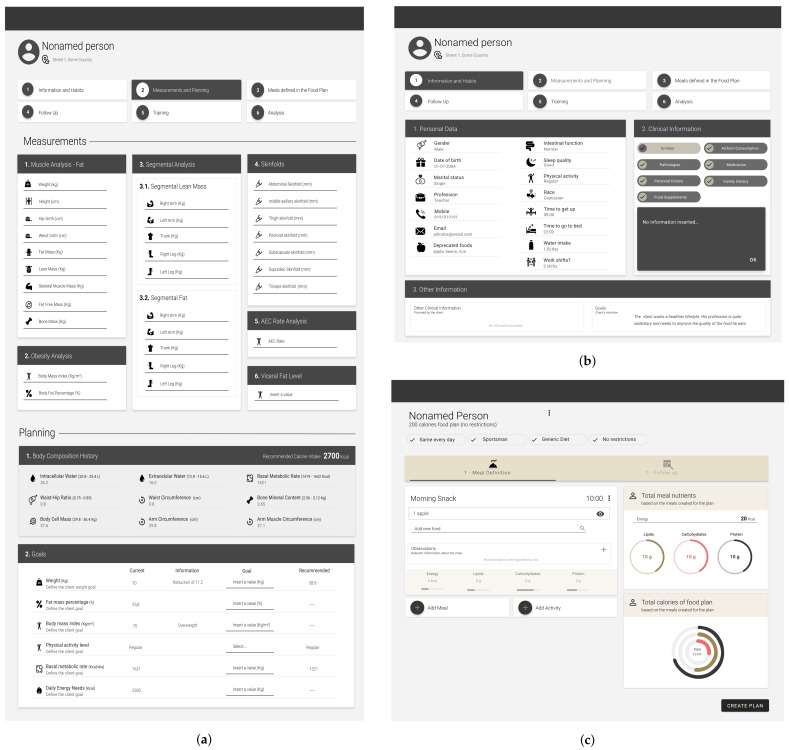
Nutritionist front-end. (**a**) Appointment. (**b**) Client details. (**c**) Food plan.

**Figure 2 sensors-22-02617-f002:**
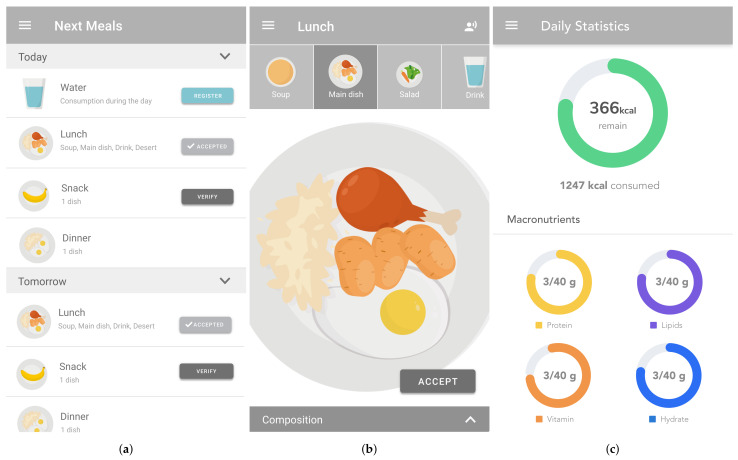
User front-end. (**a**) Daily meals. (**b**) Meal visualization. (**c**) Daily statistics.

**Figure 3 sensors-22-02617-f003:**
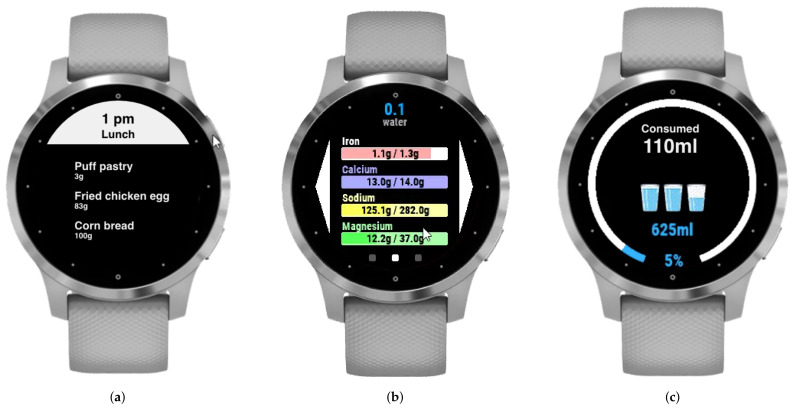
Smartwatch. (**a**) Food plan visualization and logging. (**b**) Daily control of nutrients. (**c**) Daily control of water.

**Figure 4 sensors-22-02617-f004:**
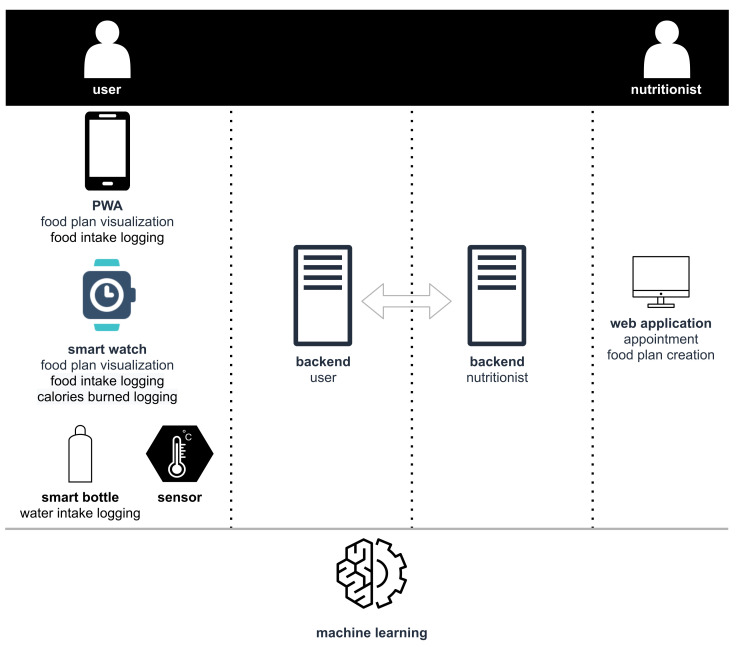
Architecture.

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
