# Peer review of "Multi-Device Nutrition Control"

_sensors, 2022, doi:10.3390/s22072617_

Round 1

Reviewer 1 Report

The manuscript present an architecture and software developed for smartphones (PWA) and smartwatches that can be used in eHealth sector.

The authors presents the concepts, requirements, and architecture of a solution that assists the nutritionist in building up and revising food plans and the user following them. The paper ends with the future work that will be will be oriented towards Alzheimer’s patients and athletes.

The paper quality good. The content is well structured, and the research is well presented, so.  

However, before publication, there are some remarks should be pointed out as follows:

  1. Line 33 - incorrect citation "[?]"
  2. Is better to mention the paper authors in text when you mention an article (e.g. line 82 "[4] explores".)

Author Response

Dear reviewer,

We greatly appreciate your thoughtful comments that helped improve the manuscript. We trust that all of your comments have been addressed.

The list of point-by-point responses:

1. Line 33 - incorrect citation "[?]"

We corrected the citation.

2. Is better to mention the paper authors in text when you mention an article (e.g. line 82 "[4] explores".)

We made the corrections suggested.

Best regards,

Carlos Cunha

Reviewer 2 Report

This manuscript presents the concepts, the involved architecture, and front-end prototypes for a nutrition control system involving Internet-of-Things, using devices as smartphones, smartwatches and smartbottles to support user to follow nutrition plans as well as nutritionists to monitoring their patients’ evolution regarding these plans. On the users' side, the system still has the premise of minimizing the interaction cost, that is, the time spent by users interacting with the system's interfaces.

I believe that the proposal is quite daring and can serve both for patients who want to benefit from specialized diets for their cases, and for professional monitoring of the evolution of these patients.

Below, I listed points that deserve attention in this work.

  • I detected a point where there is [? ] in the Introduction, third paragraph, line 33 (page 1). I consider it to be some reference that was missing. Please inform this reference.
  • In the first paragraph of Section 2, line 60 (page 2), there is a repetition about IoT: [...] such as precision nutrition, IoT, Internet of Things (IoT), web technologies, [...]. Please fix this.
  • Still in Section 2, the authors could introduce a table comparing the works related to their own proposal.
  • At the beginning of Section 3, line 112 (page 3), the authors mention: Nutrition is an eternal topic that has received more attention [...]. Here, the use of this term "eternal topic" sounded very strange, it would be better to just mention that “Nutrition is a topic that has received more attention [...]”.
  • For subsections 3.1 and 3.2, where the authors respectively list the requirements within the nutritionist and user domains, it is interesting that they demonstrate how they obtained these requirements:

a) was it through the literature? if so, which literature? (it must be mentioned in the text).

b) was it through a direct survey with nutritionists and possible users of the system? If so, how was the process done?

  • The authors did an interesting job in describing the behavior of their system, within what is possible to present in an article. However, I imagine that there are more diagrams, following the software engineering process (such as use cases, class diagrams, sequence diagrams, activity diagrams, for example). If the system is not under patent protection or computer program registration, it would be interesting to present some of these diagrams in an appendix.
  • Regarding the case studies in sections 6 and 7, I expected the authors to present data collected using the proposed system, demonstrating the benefits for both patients and nutritionists. However, only general descriptions of the case studies were presented. In its current state, the impression that remains is that the study is still in the development phase, with the two cases still in the data collection phase, and that the manuscript has the character of documenting the system and planning. If there are data collected and analyzed, it is important that they are presented, to ensure greater robustness to the study. The authors pointed out in the conclusions that these two case studies will be something for future work, which corroborates the idea that the present manuscript has a documentation and planning character.
  • How is the issue of data protection transiting through the system? How will this issue be addressed from the point of view of the user and the nutritionist? How to prevent data transiting between one end and the other from being intercepted? How to ensure that both parties make proper use of the system, preventing data leakage? These issues also need to be clarified, both from the point of view of planning the information protection and cybersecurity strategy, as well as the tools that should be used. Some suggestions for works on information security / cybersecurity, to help authors:

Sittig, D.F.; Belmont, E.; Singh, H. Improving the safety of health information technology requires shared responsibility: It is time we all step up. Healthcare 2018, 6, 7–12. https://doi.org/10.1016/j.hjdsi.2017.06.004

Alami, H.; Gagnon, M.; Ali, M.; Ahmed, A.; Fortin, J. Digital health: Cybersecurity is a value creation lever, not only a source of expenditure. Health Policy Technol. 2019, 8, 319–321. https://doi.org/10.1016/j.hlpt.2019.09.002

Poleto, T.; Carvalho, V.D.H.d.; Silva, A.L.B.d.; Clemente, T.R.N.; Silva, M.M.; Gusmão, A.P.H.d.; Costa, A.P.C.S.; Nepomuceno, T.C.C. Fuzzy Cognitive Scenario Mapping for Causes of Cybersecurity in Telehealth Services. Healthcare, vol. 9, 2021. https://doi.org/10.3390/healthcare9111504

Sondes Ksibi, Faouzi Jaidi, and Adel Bouhoula. Cyber-Risk Management within IoMT: a Context-aware Agent-based Framework for a Reliable e-Health System. The 23rd International Conference on Information Integration and Web Intelligence. Association for Computing Machinery, New York, NY, USA, 547–552, 2021. https://doi.org/10.1145/3487664.3487805

Seyed Farhad Aghili, Hamid Mala, Mohammad Shojafar, Pedro Peris-Lopez. LACO: Lightweight Three-Factor Authentication, Access Control and Ownership Transfer Scheme for E-Health Systems in IoT. Future Generation Computer Systems, Volume 96, 410-424, 2019. https://doi.org/10.1016/j.future.2019.02.020

Author Response

Dear reviewer,

We greatly appreciate your thoughtful comments that helped improve the manuscript. We trust that all of your comments have been addressed.

The list of point-by-point responses:

I detected a point where there is [? ] in the Introduction, third paragraph, line 33 (page 1). I consider it to be some reference that was missing. Please inform this reference.

We corrected the issue.

In the first paragraph of Section 2, line 60 (page 2), there is a repetition about IoT: [...] such as precision nutrition, IoT, Internet of Things (IoT), web technologies, [...]. Please fix this.

Fixed.

At the beginning of Section 3, line 112 (page 3), the authors mention: Nutrition is an eternal topicthat has received more attention [...]. Here, the use of this term "eternal topic" sounded very strange, it would be better to just mention that “Nutrition is a topic that has received more attention [...]”.

We accepted the suggestion.

For subsections 3.1 and 3.2, where the authors respectively list the requirements within the nutritionist and user domains, it is interesting that they demonstrate how they obtained these requirements.

We added the requirement analysis methodology (last paragraph before Section 3.1).

The authors did an interesting job in describing the behavior of their system, within what is possible to present in an article. However, I imagine that there are more diagrams, following the software engineering process (such as use cases, class diagrams, sequence diagrams, activity diagrams, for example). If the system is not under patent protection or computer program registration, it would be interesting to present some of these diagrams in an appendix.

We added Use Cases to the Appendix section.

Regarding the case studies in sections 6 and 7, I expected the authors to present data collected using the proposed system, demonstrating the benefits for both patients and nutritionists. 

We added the results of an analysis of interaction times using the KLM model. 

How is the issue of data protection transiting through the system? How will this issue be addressed from the point of view of the user and the nutritionist? How to prevent data transiting between one end and the other from being intercepted? How to ensure that both parties make proper use of the system, preventing data leakage?

We included Section 5.4 to describe the security aspects of our application.

Best regards,

Carlos Cunha

Reviewer 3 Report

See attached file.

Author Response

Dear reviewer,
We greatly appreciate your thoughtful comments that helped improve the manuscript. We trust that all of your comments have been addressed.

The list of point-by-point responses:

Please, avoid the use of genitive;

We did our best to avoid it.

Many references (i.e., 2, 3, 10, 11, 12, 13, 14, 16, 17, 18, 19, 20, 21, 22, 23, 30, 31 and 32) are too old. Please, substitute them with other works published not earlier than 2016;

The previous references either refer to the groundwork or are recognized by the nutrition experts (that we used as consultants) as important (e.g., calculation of body mass). Still, we agree that the article needs more recent references. We added several recent references.

The paper misses of tests. In spite of the fact that the Authors claimed to perform them in future works (i.e., lines 433-437)

To help improve the article, we added the results of an analysis of interaction times using the KLM model (Section 8). These results provide a baseline for evaluating other interaction schemes and assessing the contribution of automation (e.g., using IoT devices) to the goals established in this article. We didn't include these results before because the main objectives of the work presented are bounded by an architecture with state-of-the-art devices, the requirements, and the concepts from the nutrition domain required to develop a system with the presented characteristics.

We also included a section (5.4) describing the security concerns of the solution.

Best regards,
Carlos Cunha

Round 2

Reviewer 2 Report

The authors showed all corrections/improvements I requested before in their work, mainly with the presentation of results from an initial experiment using the e Keystroke Level Model.

For future studies, continuing what they started here, I suggest making comparisons, using statistical methods, between the results, to determine the existence (or not) of significant differences between the two scenarios considered (the Alzheimer's and the Sports' cases) for using their proposed system.

Congratulations and I hope the research continues to generate interesting results!

Author Response

Dear reviewer

Thank you very much for improving our work.

Best regards

Reviewer 3 Report

The paper notably improved after its revision. However, many genitives are still present. Please, substitute them.

Author Response

Dear reviewer
Thank you very much for helping improve our work.

We removed the genitives.

Best regards